# Knee Osteoarthritis: Epidemiology, Pathogenesis, and Mesenchymal Stem Cells: What Else Is New? An Update

**DOI:** 10.3390/ijms24076405

**Published:** 2023-03-29

**Authors:** Riccardo Giorgino, Domenico Albano, Stefano Fusco, Giuseppe M. Peretti, Laura Mangiavini, Carmelo Messina

**Affiliations:** 1IRCCS Istituto Ortopedico Galeazzi, 20161 Milan, Italy; 2Residency Program in Orthopedics and Traumatology, University of Milan, 20141 Milan, Italy; 3Postgraduate School of Diagnostic and Interventional Radiology, University of Milan, 20141 Milan, Italy; 4Department of Biomedical Sciences for Health, University of Milan, 20141 Milan, Italy

**Keywords:** osteoarthritis, knee, pathogenesis, mesenchymal stem cells, regenerative medicine

## Abstract

Osteoarthritis (OA) is a chronic disease and the most common orthopedic disorder. A vast majority of the social OA burden is related to hips and knees. The prevalence of knee OA varied across studies and such differences are reflected by the heterogeneity of data reported by studies conducted worldwide. A complete understanding of the pathogenetic mechanisms underlying this pathology is essential. The OA inflammatory process starts in the synovial membrane with the activation of the immune system, involving both humoral and cellular mediators. A crucial role in this process is played by the so-called “damage-associated molecular patterns” (DAMPs). Mesenchymal stem cells (MSCs) may be a promising option among all possible therapeutic options. However, many issues are still debated, such as the best cell source, their nature, and the right amount. Further studies are needed to clarify the remaining doubts. This review provides an overview of the most recent and relevant data on the molecular mechanism of cartilage damage in knee OA, including current therapeutic approaches in regenerative medicine.

## 1. Introduction

Osteoarthritis (OA) is a chronic disease affecting the joint and its tissues, primarily leading to progressive damage to articular cartilage and, subsequently, to the subchondral bone and surrounding synovial structures [1]. OA is a disabling condition with increasing incidence and prevalence in the general population. As one of the most common orthopedic conditions, it is associated with a high health burden and implications not only for affected patients but also for healthcare systems [2].

Different joints can be involved, yet most social OA burdens are related to hip and knee OA, both cause progressive disability and possibly lead to prosthesis replacements. Worldwide it has been recently estimated that there is an overall prevalence of about 300 million for hip and knee OA [3]. A report from the Global Burden of Disease in 2010 explored the burden of hip and knee OA comparing both conditions with other diseases [4]. According to this report, hip and knee OA represent one of the leading causes of global disability. The disease impact is measured by means of years lived with disabilities (YLDs) and disability-adjusted life years (DALYs) estimates. At a worldwide level and according to YLDs measurements, hip and knee OA is ranked as the 11th highest contributor to global disability [4].

YLDs for hip and knee OA increased from 10.5 million in 1990 to 17.1 million in 2010. Regarding disease disability, for all ages, DALYs for hip/knee OA rose from 0.42% (total DALYs in 1990) to 0.69% in 2010 [4]. Osteoarthritis is a chronic disease with multifactorial pathogenesis frequently associated with other comorbidities; therefore, disease prevention and early-stage treatment may not be easy [5]. The epidemiology and pathogenesis of hip and knee OA have been studied more extensively than other joints, and increasing interest has been devoted to the study of those molecular mechanisms, which led to cartilage damage due to their pivotal role in the pathogenesis of OA. The most frequently associated risk factors for knee OA are aging, genetic predisposition, and obesity.

The instrumental diagnosis of OA is commonly performed using conventional radiography, which can complement clinical examinations despite not always being needed for the first-line diagnosis [6]. Typically, the severity of OA in X-rays is evaluated using the Kellgren and Lawrence scale, a semi-quantitative scoring system from 0 to 4 [7]. Computed tomography (CT) and magnetic resonance imaging (MRI) represent second-line techniques, with MRI being capable of providing a full assessment of the joint (cartilage, subchondral bone, bone marrow, soft tissue structure, and inflammation status) [8]. Therefore, although MRI is an expensive examination, it has several advantages for early diagnosis of OA and research purposes [9]. Conversely, the role of ultrasound is mainly limited to soft tissue and inflammation evaluations [6]. For an early diagnosis, we report the promising use of vibroarthrography as a diagnostic tool that records vibrations caused by the joint through local accelerometers on the skin [10,11]. This technique, proposed in the literature as a safe, noninvasive, inexpensive, and reproducible tool, has demonstrated high diagnostic accuracy in the analysis of cartilage lesions [12,13,14].

This review proposes an overview of recent studies focusing on OA epidemiology, pathogenesis, and treatment. In particular, this paper will focus on the molecular mechanism in joint damage, including the most recent evidence on those mechanisms and mediators of the OA inflammatory response. Another topic in this review will be the field of regenerative medicine, in which we will discuss the most recent therapeutic approaches related to the use of mesenchymal stem cells.

## 2. Materials and Methods

Two authors (R.G. and C.M.) independently performed the literature search. The last update to the search was conducted in January 2023 and accessed PubMed, Google Scholar, Embase, and Scopus. The following keywords were used in combination with the Boolean operator AND/OR: knee osteoarthritis AND MSCs OR mesenchymal stem cells OR epidemiology OR mechanism OR pathophysiology OR pathogenesis. No additional filters or time constraints were used for the search. The same authors independently screened the resulting articles. If the title and abstract matched the topic, the full-text article was accessed. A cross reference of the bibliographies was also performed. Disagreements were debated and solved by a third author (L.M.).

## 3. Knee Osteoarthritis: Recent Trends about Epidemiology and Risk Factors

The prevalence of knee OA varied across studies according to its definition, diagnostic method, and range of age included. Such differences are reflected by the heterogeneity of data reported by studies conducted worldwide.

A recent meta-analysis conducted in China investigated the prevalence of knee OA in symptomatic patients by including 21 studies for an overall number of 74,908 subjects [15]. The overall prevalence of OA was reported to be 14.6% from studies performed in a period ranging from 2012 to 2016. Females presented with a higher knee OA prevalence than males (19.1% vs. 10.9%, respectively). The prevalence also increased with age. A demographic study from the Korean National Health and Nutrition Examination Survey (KNHANES) analyzed the prevalence of radiographic OA from 12,287 subjects older than 50 years and found it to be 35.1% [16]. Again, women were at higher risk of developing knee OA than men. Data from the Chingford Study in the United Kingdom showed that over 5 years the incidence of radiographic knee OA was 17.6% in women with ages ranging from age 45 to 64 years [17]. A study from the United States (US) estimated the incidence of symptomatic knee OA (OA) with the use of self-reported population-based data, reporting that approximately 9.29% of subjects older than 60 years presented a diagnosis of symptomatic knee OA [18]. Moreover, the overall estimated lifetime risk was reported to be 13.83%.

Although OA is not an unavoidable consequence of aging, extensive evidence confirms age as the single major risk factor for the development of OA, especially in weight-bearing joints such as hips and knees [19]. This finding has clearly emerged from several radiographic studies, which showed typical radiographic changes (joint space narrowing and osteophytosis) being very common in the population as the age increases (almost 50% of prevalence in subjects older than 80 years) [20]. At the same time, diagnosing symptomatic OA only based on radiographic findings may result in its overestimation, as considerable discordance between joint paint and imaging has been found [21].

Several risk factors are associated with knee OA, both at the personal and articular level, as highlighted by a recent article by Allen et al., who reviewed the recent evidence regarding OA [19]. The higher prevalence among females is reported by different studies, not only in the knee, but the difference in prevalence is also well-documented for differences in ethnicity. Ethnicity is a sociodemographic factor associated with variation in the prevalence of OA. As reported by different studies, the prevalence of symptomatic OA in the knee is greater in Black people than in White people [22]. The Johnston County Osteoarthritis Project reported that African Americans were more likely to have knee radiographic OA involving the tibiofemoral joint. In contrast, the prevalence of hand radiographic OA was lower compared to Caucasians [23]. Conversely, another study from the same project found low annual incidence rates of radiographic and symptomatic hip OA among African Americans, a finding that is consistent with a joint-based incidence of radiographic hip OA (compared to the person or ethnic-based model) [24]. Several studies from different countries showed that OA prevalence is higher for subjects with lower socioeconomic status, especially in hips and knees [22]. For example, Moss et al. found that lower education was associated with a higher incidence rate of radiographic hip OA, as well as lower income (< USD 15,000), even after race adjustment [24]. Moreover, for those with the lowest income, the annual incidence rate was among the highest observed (45/1000 person-years). A moderate trend of reduction in the annual incidence rates for hip OA was observed in those with the highest education, who presented a rate of 33/1000 for those with higher education compared to 40/1000 for those with lower education [24].

At a personal level, there is clear evidence that obesity represents one of the most substantial risk factors for OA, especially in the knee, with reports highlighting how obese subjects have three times the risk of developing OA [19]. While this is well documented at the tibiofemoral joint, a longitudinal study by Hart and colleagues also confirmed this at the level of the patellofemoral joint, showing that obesity is associated with increased odds of developing radiographic OA, up to 8 years later [5,25]. Moreover, the incidence of OA in the hip is significantly higher with increasing BMI: the incidence rate among obese subjects has been reported to be 48/1000 person-years, which was twice that of healthy-weight subjects (24/1000 person-years) [24]. Still, there is no clear evidence of an association between metabolic syndrome and OA at different joints, especially the hip, and knee [26]. Among the predisposing factors specific to the joint, the history of prior trauma followed by surgery has been extensively reported as an influential risk factor for OA [19], with most evidence precisely in the knee. In a recent article, Poulsen et al. performed a systematic review and meta-analysis to estimate the risk of developing OA after a knee injury [27]. The paper, which included more than 50 studies for a total of about 1 million subjects evaluated, reported the following odds of developing OA: four times higher after an anterior cruciate ligament tear (ACL) and six times higher after the combination of ACL and meniscal tear. Further, lower limb imbalance has been reported as a specific joint risk factor, with evidence of an increased risk of OA by radiographic findings and symptoms [18]. A study by Kim et al., in 2018, evaluated the prevalence of hip OA by assessing lower limb leg inequality on radiographs from the Multicenter Arthritis Study (MOST) and the Osteoarthritis Initiative (OAI) for a total of 1966 and 2627 subjects, respectively. For a leg length inequality (LLI) ≥ 1 cm, the odds ratio for developing incident hip OA in the shorter leg was 1.39, yet the odds ratio was more than double (4.20) for a LLI ≥ 2 cm. The authors, therefore, concluded that having a LLI of at least 2 cm is a risk factor for developing hip OA, while the shorter leg is at a higher risk [28]. Similarly, in the hip, Moss et al. reported the highest incidence rates of OA were in those with previous injuries [24]. Static alignment is another joint-specific risk factor that has been traditionally considered a strong predictor of knee OA, particularly frontal plane knee alignment [29]. Still, the recent data regarding the association between static alignment and knee OA remains mixed: a recent meta-analysis highlighted that similar odds of varus and valgus malalignment were observed in adults with prevalent knee OA [30]. Nevertheless, results from the Wuchuan OA Study (a population-based longitudinal study of risk factors for knee OA in rural China) observed that varus alignment was associated with the prevalence of medial compartment knee OA (with an OR of 6.1). In contrast, valgus alignment was associated with lateral compartment knee OA (OR = 5.0) [31]. Regarding sports participation, a moderate to strong association with increased risk for developing OA at the hip and knee has been reported for those activities that include high load and potential risk for trauma to the weight-bearing joints (e.g., high-impact sports). Driban et al. performed a systematic review to evaluate the association between participation in specific sports and knee OA [32,33]. Overall, the reported prevalence of knee OA in sports participants was found to be 7.7% compared with a prevalence of 7.3% among the control group of nonparticipants. Certain sports were associated with a higher prevalence of knee OA, such as long-distance running at the elite level (OR = 3.3), soccer (OR = 3.5), weightlifting at a competitive level (OR = 3.5), and wrestling (OR = 3.8) [32]. Another systematic review investigated the association of high-impact sporting activities with the risk of developing hip OA, showing that handball was associated with the highest rate of OA prevalence (five times higher than matched controls). As for knee OA, soccer participants also presented with a higher risk of hip OA (between two and nine times, according to the X-ray diagnostic definition) [34]. In the hip, OA risk may also be associated with femur-acetabular impingement (especially the CAM type), a condition that can be manifested in sporty adolescents [1].

## 4. Pathogenesis of OA

At the joint level, many knee structures offer a certain degree of mechanical and functional support to maintain a healthy joint [35]. The subchondral bone (mainly composed of mineralized type I collagen) assists the articular cartilage (mainly constituted of type II collagen and proteoglycans) in providing a surface for joint movement. The menisci provide a significant role in attenuating mechanical forces due to their structure of water, proteoglycans, and collagen. Finally, the synovial fluid to lubricate the articular space is produced by the synovial membrane: it is generally composed of hyaluronic acid and lubricin (also known as proteoglycan 4, PRG4) [36]. Of note, the superficial cartilage layer plays a vital role in constricting water content within the cartilage, acting as a regulator of water content: injuries of early damage to the superficial layer change the water content and, therefore, diminish the load-bearing properties of cartilage [37].

In the early stages of knee OA, alterations in the structure of collagen and proteoglycans are observed, together with degenerative changes in the meniscal structure. Both of these conditions lead to overcoming the compensatory mechanisms that limit the articular cartilage damage, ultimately, causing meniscal damage and articular cartilage erosions [38]. Of note, the proinflammatory role of cytokines has been confirmed as an essential mechanism of articular damage during the early stages of OA. In response to cartilage erosions, chondrocytes first go through a phase of hypertrophic activity to increase the matrix synthesis, producing inflammatory mediators that propagate cartilage degradation [39]. The ultimate stage of cartilage destruction is chondrocyte apoptosis, leading to an imbalance in the synthesis and catabolism of collagen and proteoglycans in favor of catabolism. Inflammatory mediators spread to other joint structures, causing changes in the synovial tissue and subchondral bone causing bone sclerosis and increasing the thickness of the synovial membrane and capsular structures [40]. Ultimately, gaps are produced in the cartilage surface, with free cartilage fragments propagating the synovial inflammatory condition, a condition that further decreases the synthesis of synovial molecules (lubricin and hyaluronic acid) [40]. The expression of type II collagen (a prominent component of cartilage) decreases during chondrocyte growth; therefore, mature chondrocytes are incapable of producing type II collagen de novo.

Of note, although the changes in the subchondral bone have been traditionally implicated in the OA pathogenesis at a later stage, recent studies identified this joint structure as one of the first actors in the pathological process. Indeed, recent evidence suggests the existence of specific crosstalk between subchondral bone and articular cartilage. For example, contributing to OA progression has been attributed to altered venous outflow circulation in the subchondral bone, causing physicochemical changes that stimulate osteoblasts to express bone remodeling and cartilage-damaging cytokines [41]. A recent study by Wu et al. investigated the relationship between OA subchondral bone-derived exosomes (membrane-derived vesicles implicated in intercellular communication) and chondrocytes. The study showed that in coculture studies, OA sclerotic subchondral bone osteoblast-derived exosomes were internalized by chondrocytes, causing the expression of catabolic genes, as observed in OA cartilage. Therefore, it seems that such exosomes are crucial in OA progression, representing a possible target for OA treatments [42]. As osteoarthrosis is a degenerative disease, the concept of cellular senescence has also been suggested to understand its pathogenesis [43]. Cellular senescence of chondrocytes has been associated with the progressive reduction in cell cycle activity until it eventually stops, apoptosis resistance, and the progressive production of senescence-associated secretory phenotype (SASP) [43]. Different types of molecules can represent SASPs, yet—in the setting of OA—SASPs are all those inflammatory factors previously described (such as cytokines and chemokines). SASPs can be produced not only by chondrocytes, but also by other cells within the OA joint, such as osteoblasts, synovial fibroblasts, and macrophages [44]. For example, TGF-β and IL-6 are both SASP factors that can contribute to chondrocyte aging by activating p15, p21, and p27, therefore, promoting its senescence through the SMAD complex or STAT3 pathway [45,46]. Additionally, the release of SASPs by senescent chondrocytes can produce a chemotactic effect on surrounding immune cells; thus, establishing an inflammatory environment further stimulates cartilage degradation [43].

## 5. The OA Inflammatory Response: Mechanism and Mediators

The OA inflammatory process starts in the synovial membrane with the activation of the immune system, involving both humoral and cellular mediators [47]. A crucial role in this process is played by the so-called “damage-associated molecular patterns” (DAMPs), which represent small fragments produced during cartilage degradation that are released into the joint space from the extracellular matrix (ECM) [48]. DAMPs stimulate the production of inflammatory mediators by fibroblasts and macrophages of the synovium, such as chemokines and cytokines. As a response, articular cartilage chondrocytes produce metalloproteinase, leading to a vicious damaging circle between the synovial membrane and cartilage [49].

DAMPs are endogenous molecular products and can be divided into “intracellular” or “extracellular”. Intracellular DAMPs are represented by several molecules deriving from apoptotic, while extracellular DAMPs are represented by the ECM components themselves (i.e., proteoglycans or glycoproteins). DAMPs can activate different types of receptors, known as pattern recognition receptors (PRRs), which are abundantly present on the cell surface of synoviocytes and chondrocytes, among others. Examples of PRRs are the receptor for advanced glycosylation end products (RAGEs), a member of the immunoglobulin superfamily, or the Toll-like receptors (TLRs) and NOD-like receptors (NLRs) [50]. Once DAMPs are bound to PRRs, a signaling cascade, ultimately, causes the activation of crucial regulator factors for the inflammatory response, one of them being the nuclear factor-kB (NF-kB) [51]. As a consequence, several inflammatory mediators were produced: cytokines such as the interleukin (IL)-1b and IL-6 or the tumor necrosis factor (TNF)-a, cathepsins, chemokines, and catabolic factors, such as matrix metalloproteinase (MMP), as well as the complement cascade [52] (Figure 1).

### 5.1. An Insight into ECM-Related DAMPs

The damage to the ECM caused by the activity of MMPs can produce several fragments of molecules working as DAMPs that can promote inflammation, thereby progressing the process of cartilage disruption [21]. The proposed vicious circle between synovial membrane and cartilage starts with DAMPs production from ECM, causing the exposure of hidden epitopes and ligand receptors on the ECM surface, followed by interaction with inflammatory mediators that further stimulate cartilage degradation through specific enzymes [22]. A schematic representation of the mechanisms and implications of different ECM-related DAMPs molecules in the generation of inflammatory response in OA is shown in Figure 2.

Fibronectin is an ECM glycoprotein with a high molecular weight (~500–600 kDa) that can be damaged during cartilage degradation through proteolytic cleavage, causing the production of fibronectin fragments stimulating the chondrolytic process [23]. This mechanism is sustained through the increase of cytokine production and the inhibition of proteoglycan synthesis, and a stimulus to increase the expression of MMP. As an example, a specific amino-terminal fibronectin fragment was shown to be able to stimulate the production of inflammatory cytokines in human cartilage cultures (such as IL-1a, IL-1b, and TNF-a), as well as MMP-1 and MMP-3 [17,24]. Fibronectin fragments are also capable of upregulating the Toll-like receptor 2 (TLR-2), via an interleukin-1 receptor antagonist protein (IL-1ra), further supporting the synthesis of proinflammatory cytokines [25].

Another vital stimulus that can activate humoral immunity is the activity of peptides derived from type II collagen. It has been shown that the incubation of human chondrocytes with collagen II peptides caused a dose-dependent induction of several proinflammatory MMPs (such as MMP1, MMP3, MMP13, and MMP14) and cytokines (IL-1beta, IL-6, and IL-8) [26]. This contributes to the vicious circle of cartilage damage, as proinflammatory molecules production, in turn, contributes to the further release of collagen II fragments from mature collagen II fibers. Another type II collagen peptide that has been studied in human chondrocytes is the N-terminal fragment (29-mer fragment), which has been shown to enhance the expression of several cathepsins (B, K, and L) at the mRNA, protein, and activity levels; this induction is related to the activation of protein kinase C and p38 MAP kinase [17]. A study by Lambert et al. highlighted the role of Coll2-1 (a synthetic peptide of type II collagen) as a promoter of cartilage degradation by stimulating IL-8 production by synoviocytes and MMP-3 production by chondrocytes [27]. In a culture of human chondrocytes, Coll2-1 also significantly affected vascular endothelium growth factor (VEGF) gene expression, ultimately causing synovitis and cartilage damage through the loss of proteoglycans, and subchondral bone remodeling [27].

Hyaluronic acid (HA), a nonsulfated glycosaminoglycan, is another ECM component widely present in the synovial fluid. HA is widely used to treat knee OA in the form of exogenous HA (usually with an injection of high molecular weight HA). Interestingly, it has been shown that a very low molecular weight of HA (resulting from the degradation of bigger HA molecules) can stimulate inflammation by inducing MMP and nitric oxide production [28]. Another study focused on the role of hyaluronan released in the joint space after traumatic injuries, showing that it can work as an inflammatory trigger of chemokine release and as a signal of cartilaginous trauma [29]. In their study, Yamasaki et al. investigated the role of NLRP3/cryopyrin, which represents a component of the inflammasome that contributes to the inflammatory response in sterile inflammation after trauma, under the stimulus of short HA oligosaccharides (4–18-mer, 0.777–3.5 kDa). Such small HA oligosaccharides activated the inflammasome, via the NLRP3/cryopyrin to activate and release IL-1 [29].

Another ECM molecule acting as DAMP is lubricin, also known as proteoglycan 4 (PRG4). Lubricin is a mucin-like glycoprotein secreted by synovial fibroblasts and superficial zone chondrocytes and is present in the articular cartilage, working as a joint lubricant [30]. It has been shown that a reduction in the expression of PRG4 is associated with OA progression and results in premature joint damage [17]. At the same time, lubricin can also play a role in regulating the inflammatory response through interaction with CD44 and regulation of the activity of synovial fibroblasts. The role of lubricin has been shown in vitro, where it was shown to bind to and regulate the activity of the TLRs (TLR-2, -4, and -5), modulating the secretion of chemokines and cytokines [31]. A very recent study by Huang et al. showed that in the setting of OA, truncated glycans of lubricin stimulate the synoviocyte secretion of VEGF, IL-8, and macrophage inflammatory protein 1-alpha (MIP-1α) as a proinflammatory cytokine, exacerbating the synovial inflammatory condition [32].

Tenascin-C (TNC) is another ECM component whose role as a proinflammatory mediator in OA has emerged in recent years [33]. TNC is a large glycoprotein of the ECM consisting of four domains and whose expression in the joint depends on the developmental stage of the articular cartilage; during the chondrocyte maturation, the expression of TNC progressively reduces, nearly disappearing in the mature joint cartilage in adults [34]. Nevertheless, TNC expression is upregulated in diseased joints, such as OA subjects and rheumatoid arthritis [33]. The presence of TNC fragments in the cartilage of OA subjects was demonstrated, as well as its role in promoting cartilage degradation by inducing aggrecanase activity. In particular, recombinant TNC fragments (the Fn type III domains 3–8 of TNC and the epidermal growth factor-like) mediate cartilage degradation through the induction of aggrecanase activity [35]. TNC has been shown to interact with TLR-4 as an endogenous inflammatory mediator, with up to three different sites of interaction within the C-terminal fibrinogen-like globe (FBG) of TNC [36].

Other DAMPs are a stimulus for the induction of inflammatory responses during OA. Bone sialoprotein I (BSP-1), an ECM non-collagenous protein, is expressed by different cells, such as the synoviocytes, chondrocytes, and activated immunity cells [17]. Compared to healthy subjects, an increased level of BSP-1 was reported in the articular cartilage and synovial fluid of patients with OA, and its levels are also related to the grade of OA [37]. Similarly, elevated levels of cartilage oligomeric matrix protein (COMP) have been reported in the synovial fluid of OA patients. This ECM protein, also known as thrombospondin-5, is abundantly present in the articular cartilage, representing a marker of cartilage turnover [38]. In a recent study on knee OA performed in 150 subjects (100 cases and 50 controls), a statistically increased level of serum COMP was found in subjects with grades I and II Kellgren–Lawrence OA compared to healthy subjects [39]. Byglican is a small leucine-rich repeat proteoglycans (SLRP) family member, together with other proteins, such as fibromodulin, lumican, decorin, osteoadherin, and chondroadherin. SLRPs are constituents of the “core matrisome” of ECM, generally composed of collagens, proteoglycans, and glycoproteins. The leading role of the SLRPs is regulating collagen fibrillogenesis, which is the assembly of ECM components [35]. Biglycan is released from ECM after tissue injury, acting as a DAMP and ligand for TLR-2/4 [40]. The role of biglycan was studied in an experimental model, showing that the exposition of chondrocytes to high levels of biglycan results in the expression of numerous inflammatory mediators, such as IL-1β, IL-6, IL-17, and MMP-13 [40].

### 5.2. Intracellular Molecules Acting as DAMPs

Intracellular DAMPs are represented by different molecules with immunogenic properties produced by disrupting apoptotic or necrotic cells [48]. An association has been found between fibrin deposits and the severity of cartilage damage [52]. The articular damage is exerted through the upregulation of Adamts5 and MMP-13 in chondrocytes and the development of chondro-synovial adhesion, which specifically occurred in fibrin-rich cartilage areas [52]. Fibrin deposits were also found to stimulate the production of inflammatory cytokines by macrophages, via the TLR-4 pathway [48].

Alarmin molecules, such as the S100A8 and S100A9, are locally released by the synovial cells during inflammatory OA, as well as by monocytes, activated macrophages, and neutrophils [48]. Additionally, in OA subjects, an association was found between the expression of S100A8 and S100A9 alarmins and the expression of MMPs, such as MMP-1 and MMP-3, as well as with proteoglycan depletion [53]. A recent article where experimental mice with OA mice showed that S100A8 and S100A9 increased the mobilization of proinflammatory monocytes from the BM to the synovium [53]. Another well-known proinflammatory alarmin involved in the development of knee OA is the high-mobility group box 1 (HMGB-1) [54]. In their study, Ke et al. evaluated the amount of HMGB-1 in the synovium of OA subjects, comparing it with controls, and assessed the association between synovial fluid HMGB-1 levels and the severity of synovitis [54]. They demonstrated that HMGB-1 levels were higher in OA patients both in the synovium and synovial fluid and that synovial fluid HMGB-1 levels were positively associated with the degree of synovitis.

Regarding plasma proteins such as alpha1- and alpha2-macroglobulin, both have been traditionally implicated in the inflammatory mechanism of OA as they can induce the production of cytokines by macrophages through the TLR4 [48]. However, more recent studies have found that alpha 2-macroglobulin can inhibit the IL-1β—NF-κB inflammatory pathway [55]. In their study, Sun et al. demonstrated using immunoprecipitation that alpha2-macroglobulin is capable of binding IL-1β. At the same time, using Western blot analysis, alpha2-macroglobulin neutralized both IL-1β and NF-κB, showing the capability to directly neutralize IL-1β and downregulate the NF-kB inflammatory response. Additionally, alpha2-macroglobulin decreased the levels of MMPs [55]. Finally, several types of microcrystals that can be found in joint disorders can act as a trigger for inflammation response. For example, calcium pyrophosphate dehydrate (CPPD) crystals have been shown to stimulate the production of inflammatory cytokines, MMPs, and prostaglandins by both synoviocytes and chondrocytes [48]. Uric acid is also known to represent a harmful crystal for joints, as confirmed by Eleftheriadis and colleagues; their study showed how urate crystals could directly activate the T-cell receptor (TCR) complex, inducing the proliferation of T-cells [56].

### 5.3. The Role of Cellular Receptors in DAMPs Mediated Activity

The inflammatory activity of DAMPs molecules takes place through several receptors expressed by different cells, mainly the TLRs, the RAGE receptors, and the NOD-like receptors (NLRs) [48]. TLRs are type I transmembrane receptors with up to eleven TLRs that have been identified in human cells: their inflammatory activity is exerted through the activation of nuclear factors (e.g., NF-kB), the recruitment of myeloid differentiation primary response 88 (MyD88, a protein acting as adapter by connecting signaling proteins from outside to inside the cells), the adapter protein known as TRIF (TIR domain-containing adaptor-inducing interferon), and by initiating the activation pathways of phosphoinositide 3-kinases (PI3K) and mitogen-activated protein kinase (MAPK) enzymes. Ultimately, these pathways regulate gene transcription and translation of proinflammatory cytokines [57]. Among the different TLRs, TLR-2 and TLR-4 are those mainly involved in the OA inflammatory response. They are expressed by several joint cells such as synoviocytes, chondrocytes, monocytes, and other immune cells [48]. More in detail, TLR-4 activation increases the expression of MMPs, the release of NO, and the production of IL-1b, while reducing type II collagen synthesis [58]. TLRs also may play a key role in the innate immune response by bonding and stimulation with pathogen-associated molecular patterns (PAMPs), which are molecules that may derive from microorganisms that are produced in the setting of microbial infection [57]. A recent study showed that lumican, a major extracellular matrix glycoprotein that is present in adult cartilage, is upregulated in the synovial fluid of OA patients. In their study, the authors report that elevated levels of lumican stimulate PAMP-induced lipopolysaccharide TLR4 activation, causing the expression of proinflammatory molecules and leading to cartilage degradation [59]. NLRs are intracellular sensors that can be stimulated both by DAMPs molecules as well as by PAMPs pathogen-associated molecular patterns (PAMPs). NLRP3, one of the most studied receptors of this family, is expressed in several joint cells. When activated, it is able to interact with different proteins to create “the inflammasome”, a multimeric structure that induces the production of different cytokines [60]. Another bond for DAMPs molecules is the RAGE transmembrane receptor, whose interaction is mediated by its extracellular part, whereby it is connected with the intracellular cytoplasmatic domain through a transmembrane domain. RAGE activation is a positive stimulus for the MAPK and NF-kB pathways, causing the expression of proinflammatory cytokines [61].

### 5.4. The Actors of Cellular Response in OA

Several types of cells play a pivotal role in mediating the inflammatory response to OA, and these cells can be both residential in the joint as well as derived from the immune system [35]. Osteoclasts can be involved in the process of cartilage degradation by releasing different inflammatory mediators, such as different types of cytokines (IL-1b, IL-6, and TNF-alpha), the vascular endothelial growth factor (VEGF), and chemokines. All these mediators are involved in mechanisms related to the increase in the vascularization of the synovial lining, which in turn causes the migration of blood monocytes that, within the synovial space, differentiate into macrophages [62].

Among the immune system cells, neutrophils play a major role in cartilage damage and degradation. In fact, neutrophils are widely present in the inflammatory synovial fluid, producing several mediators capable of damaging cartilage [63]. A recent study demonstrated that in subjects suffering from knee OA, the abundance of neutrophils in the synovial fluid is associated with elevated cartilage-damaging mediators, such as TNF-α, IL-1RA, MMP-9, and VILIP-1 [63]. Additionally, reactive oxygen species (ROS) formed by neutrophils can stimulate proteolytic enzyme production, representing the main actors of cartilage degradation [64].

Monocytes are also involved in OA pathogenesis, as they can activate the subchondral bone and synovium by releasing several cytokines and chemokines. These mediators are found in high concentrations within the synovial fluid of OA patients [65]. As shown in animal studies, a reduction in monocyte activation (with consequent depletion of synovial macrophages, which derives from monocytes) may be beneficial for OA by decreasing cartilage destruction [66]. Similarly, macrophages are also considered to play a pivotal role in influencing the course of OA. They can act in proinflammatory and anti-inflammatory ways, according to their phenotype polarization (M1 or M2). M1 macrophages are proinflammatory cells capable of producing inflammatory mediators. At the same time, the M2 phenotype is considered chondroprotective as it can inhibit chondrocyte apoptosis and increase the expression levels of prochondrogenic TGF-β1 mRNA. OA progression has been shown to be closely related to the imbalances between the M1/M2 phenotype of synovial macrophages. Therefore the inhibition of M1 polarization to the advantage of the M2 phenotype represents a possible strategy for slowing the progression of OA [67].

## 6. Regenerative Medicine for Knee OA: Mesenchymal Stem Cells (MSCs)

OA is a multifactorial condition, followed by an individual response to therapy, which differs from patient to patient [68]. There is an intrinsic reactive capacity of cartilage in the face of defects, although it is limited. Cartilage tries to respond by producing cartilage-like tissue, which tends to be fibrous or fibrocartilaginous. This tissue is biomechanically and biochemically different [69]. Here is where MSCs come in. Assuming they can make up for this intrinsic regenerative capacity of cartilage, they have already demonstrated their benefits in the knee joint affected by OA [70,71]. MSCs are multipotent cells derived from a population of adult stem cells that can be isolated from numerous tissues [72,73]. MSCs are pluripotent progenitor cells derived from a population of adult stem cells that can be isolated from numerous tissues [74]. The definition and origin of MSCs are still debated, as some report a perivascular origin [75]. Credentials derive from the demonstrated capabilities of MSCs for multidirectional differentiation and the ability to modulate immunity and inflammation and improve angiogenesis, cell survival, and differentiation [76,77,78]. A vital aspect to consider lies in the interpretation and understanding of the actual mechanism of action of MSCs [79]. From observing some studies, it was concluded that these cells behave as a stimulation tool through growth factors, mediators of tissue repair, rather than having a replacement action [75]. Moreover, another advantage lies in the fact that intra-articular injection of MSCs is undoubtedly a much less invasive alternative to other surgical treatments [80].

These cells can be isolated from numerous tissues in the body and have different characteristics depending on the tissue of origin [81,82]. In particular, synovium-derived MSCs have been shown to have better chondrogenesis potential than those derived from other tissues [83]. The key may lie in the expression of growth differentiation factor 5 (GDF5), which is shared with articular cartilage [84]. However, problems persist related to the digestion of enzymes to isolate these cells [85]. Numerous studies have already reported clinical trials for intra-articular injection of MSCs in patients with OA, confirming that it led to improved clinical and radiological parameters and was a safe procedure [71,86,87].

## 7. Recent and Innovative Therapeutic Approaches of MSCs

Another exciting and promising source of MSCs effective in the treatment of OA is found in the infrapatellar fat pad, which has been shown to have better chondrogenic potential than those derived from bone marrow or subcutaneous fat [88]. This site and the synovial site mentioned earlier are widely used in older patients [89]. Currently, bone marrow-derived stromal cells are the most common clinical source of MSCs [90]. These are harvested from a bone marrow containing hematopoietic stem cells, endothelial progenitor cells, and related cytokines and growth factors [91]. Other researchers have used adipose tissue-derived stromal cells as an alternative cell line [92]. There is still debate between these two sources of MSCs [93]. Adipose tissue has a higher stem cell yield [94,95]. In addition, cells from adipose tissue show greater longevity than those derived from BM [96]. However, at the expense of a lower ability to differentiate into osteocartilaginous tissue [97,98]. On the other hand, other studies support the supremacy in terms of efficacy and safety of adipose tissue MSCs for the treatment of OA [99]. There is undoubtedly conflicting evidence in the literature [100]. The exciting study by Muthu et al. analyzed the difference between these two cell populations [101]. The analysis showed that at a 1-year follow-up, MSCs from adipose tissue were superior in their safety and consistent efficacy in improving pain, functional, and radiological outcomes.

The number of MSCs to be injected to achieve a significant therapeutic effect is still debated. For autologous MSCs, it is not established that higher dose intra-articular injections could necessarily result in better therapeutic effects [80]. On the contrary, one study reports a more significant reduction in pain symptoms and a higher functional score for a lower dose of MSCs [102]. Other studies have investigated the relationship between the number of MSCs to be injected and functional outcomes, with different follow-ups, leading to conflicting results on which amount is more effective [103,104].

Another debate in the literature concerns the nature of MSCs, comparing allogeneic versus autologous sources. Both demonstrated improvement in pain symptoms after six months, but after one year of follow-up, autologous sources proved superior in terms of efficacy, functional outcomes, and safety [105]. However, although the results of autologous MSCs seem better, it is fair to reflect on the residual regenerative potential of these cells. The study by van Rhijn-Brouwer et al. showed that this potential is influenced by the patient’s cardiovascular comorbidities [106]. This is a finding that should not be underestimated because patients with OA are generally elderly with a fair number of comorbidities.

Recently, a critical therapeutic role complementary to the activity of MSCs belongs to exosomes [107,108]. Paracrine secretion of these specific extracellular vesicles may play a role in limb tissue repair, inhibiting OA development [109,110]. Through the transport of specific substances, these vesicles act as mediators of intercellular communication, generating a response in the cell [111]. The mechanism of function is accomplished by regulating immune reactivity and inhibiting apoptosis [112,113]. The future of MSC-based therapies is definitely characterized by these vesicles, which could optimize the benefits of MSCs [114].

## 8. Conclusions

OA is the most common orthopedic disorder and increasing interest has been devoted to studying those molecular mechanisms fundamental in pathogenesis. A consistent part of the social burden of OA is related to knee OA. It is crucial to understand that OA is the end stage of numerous alterations that overcome the compensatory mechanisms, which limit articular cartilage damage, as the reactive capacity of cartilage tissue in the face of injury is limited. In this, inflammatory processes and immune system activation play a crucial role. MSCs may be a promising therapeutic option because of their demonstrated regenerative capabilities. Many issues are still debated, including the best source of MSCs and the potential role of exosomes. Further clinical and in vitro studies are needed to clarify the remaining doubts.

## Figures and Tables

**Figure 1 ijms-24-06405-f001:**
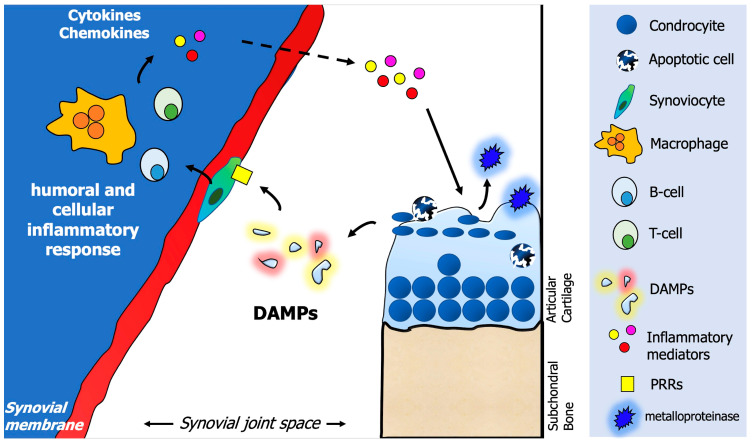
The vicious cycle between synovial membrane and cartilage and the crucial role of “damage-associated molecular patterns” (DAMPs). From the activation of pattern recognition receptors (PRRs) by DAMPs, the cascade of activation of the inflammatory response by synovial fibroblasts and macrophages is initiated. The vicious cycle is sustained by the damage to the ECM caused by the activity of MMPs, which further produce several fragments of molecules working as DAMPs.

**Figure 2 ijms-24-06405-f002:**
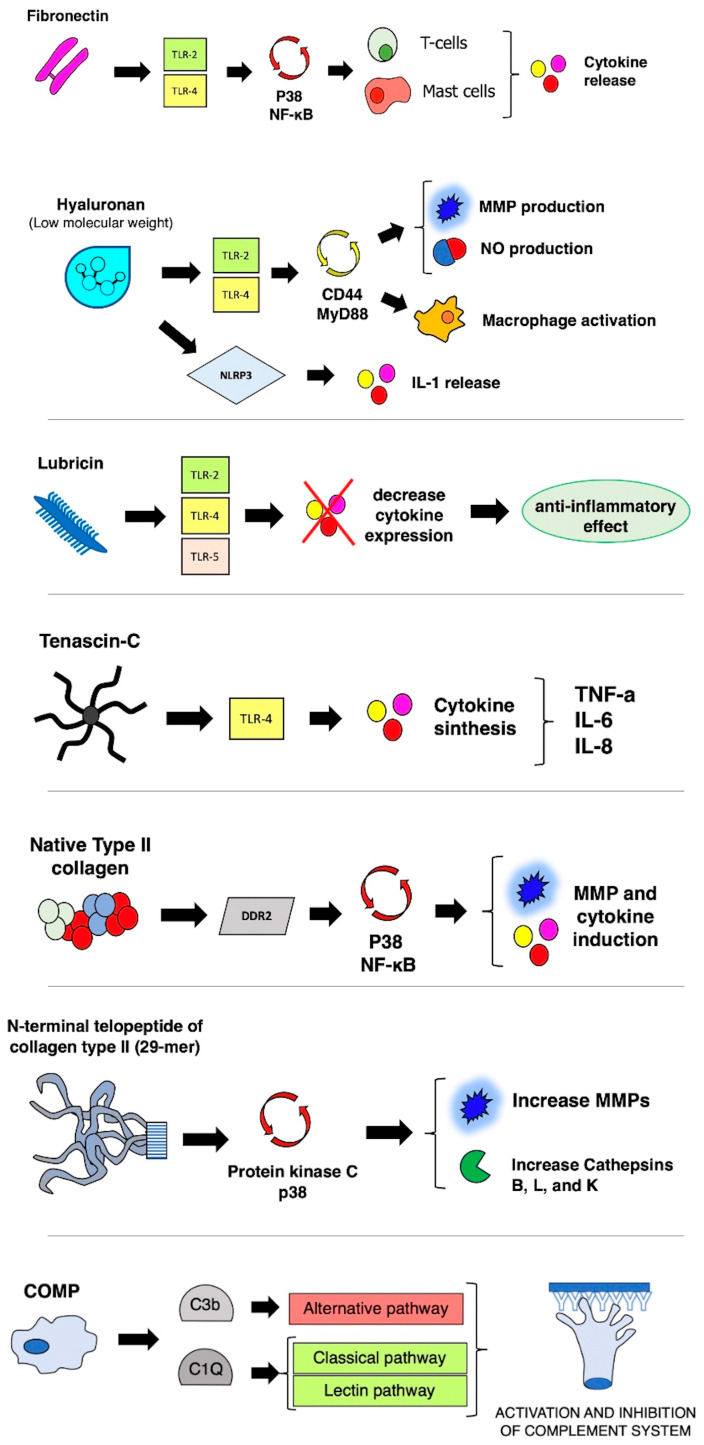
Schematic representation of inflammatory mechanisms mediated by different DAMPs. COMP: cartilage oligomeric matrix protein. Receptors: TLR: Toll-like receptor; NLRP3: NOD-like receptor family, pyrin domain containing 3; DDR2: Discoidin domain-containing receptor 2. Signaling pathway: p38: p38 mitogen-activated protein kinases; NF-kB: nuclear factor-kB; MyD88: myeloid differentiation primary response 88. Effects: MMP: matrix metalloproteinase; NO: nitric oxide; IL: interleukin; TNF: tumor necrosis factor.

## Data Availability

Not applicable.

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
