# Peer review of "Knee Osteoarthritis: Epidemiology, Pathogenesis, and Mesenchymal Stem Cells: What Else Is New? An Update"

_ijms, 2023, doi:10.3390/ijms24076405_

Round 1

Reviewer 1 Report

This review certainly highlights a number of questions about the pathogenesis and possible treatment of osteoarthritis. The potential sources of inflammation and the role of both chondrocytes and synoviocytes in its development are well covered. At the same time, I would like to draw the authors' attention to a number of remarks.

The concept of the review is not entirely clear, and why these sections have been taken into consideration if, for example, the introduction identifies age, genetics and obesity as the main components of pathogenesis. In the pathogenesis section, the inflammatory component is covered fragmentarily, the role of the cellular immune system is not disclosed, in the treatment section the emphasis is on stem cells, although it would be logical to present information on the therapy of low-level inflammation caused by the interaction of DAMPs and joint structures - optionally - doi: 10.5483/BMBRep.2020.53.2.291 and similar.

No data are given for currently registered and completed clinical trials, in particular from clinicaltrials.gov.

The article points out the secondary role of the subchondral bone in the development of osteoarthritis, although several studies have identified this joint structure as the first link in the pathological process (DOI: 10.3390/cells10020251, DOI: 10.1007/s11926-017-0660-x).

In the epidemiology section, ethnic differences in the prevalence of the disease should be highlighted in more detail, as data from patients from Southeast Asia are mainly presented.

To summarise:

- Outline the choice of concept in the introduction

- Supplement the epidemiology data

- Extend the material on inflammation, add data on protein-associated molecular patterns (PAMPs), possibly senescence-associated secretory phenotype (SASP) of chondrocytes and osteoblasts

- Describe possible treatments for low-level inflammation, provide data on clinical trials

Author Response

We really thank the reviewer for his/her comments. Please find attached our point to point response, we hope the revisions were satisfactory.

Reviewer: In the pathogenesis section, the inflammatory component is covered fragmentarily, the role of the cellular immune system is not disclosed, in the treatment section the emphasis is on stem cells, although it would be logical to present information on the therapy of low-level inflammation caused by the interaction of DAMPs and joint structures - optionally - doi: 10.5483/BMBRep.2020.53.2.291 and similar.

Response: We thank the reviewer for his/her suggestions. A specific paragraph regarding the role of cellular immune system was added with the title “The actors of Cellular Response in OA”. Just above it, an additional brief paragraph discussing “The role of cellular receptors in DAMPs mediated activity” was added to the pathogenesis section to further discuss the interaction between DAMPs and immune/joint cells. Regarding, possible treatments for low-level inflammation caused by the interaction of DAMPs and joint structures, we preferred not to include the in the discussion of this article. We preferred to focus this review on the latest updates on a few aspects we selected and considered "hot topics," such as, for example, epidemiology and the critical consequences derived, pathogenesis, and utilization of MSCs as therapy.

Reviewer: No data are given for currently registered and completed clinical trials, in particular from clinicaltrials.gov.

Response: Thanks for your advice. Some of the most significant clinical trials are reported in the section “Regenerative medicine for knee OA: Mesenchymal Stem Cells (MSCs)”. General results have been added in that sentence. (lines 574-576).

Reviewer: the article points out the secondary role of the subchondral bone in the development of osteoarthritis, although several studies have identified this joint structure as the first link in the pathological process (DOI: 10.3390/cells10020251, DOI: 10.1007/s11926-017-0660-x).

Response: we thank the reviewer for pointing out this important aspect, as well for the interesting papers suggested. According to your suggestions, we added a specific paragraph regarding the role of subchondral bon

Reviewer: In the epidemiology section, ethnic differences in the prevalence of the disease should be highlighted in more detail, as data from patients from Southeast Asia are mainly presented.

Response: we expanded the epidemiology section with a dedicated paragraph regarding ethnic difference, together with other sociodemographic factors associated with OA

Reviewer: Outline the choice of concept in the introduction

Response: Introduction was expanded, and we clearly defined the purpose of our paper.

Reviewer: Supplement the epidemiology data

Response: We supplemented the epidemiology section by expanding the importance of ethnicity and other sociodemographic factors (education, money, income), and we also expanded the part related to obesity as a risk factor.

Reviewer: Extend the material on inflammation, add data on protein-associated molecular patterns (PAMPs), possibly senescence-associated secretory phenotype (SASP) of chondrocytes and osteoblasts

Response: we extended the part on inflammation by adding data about PAMPs. We also added a paragraph about SASPs at the end of the paragraph “Pathogenesis of OA”

Reviewer: Describe possible treatments for low-level inflammation, provide data on clinical trials

Response: see the above answer.

Reviewer 2 Report

The review article by Giorgino et al, focuses on knee osteoarthiritis (OA) and the current status of its epidemiology, pathogenesis and how mesenchymal stem cells could play a key role in its treatment therapy. This article touches upon inflammation that leads to activation of immune system via damage-associated molecular patterns (DAMP) and pattern recognition receptors (PRRs) axis. It’s a well written document covering latest literature, however there are certain areas in the paper that need attention.

Here are some of my suggestions that could help to improve the manuscript:

1.       In the abstract the authors have written “Mesenchymal stem cells (MSCs) may be a promising therapeutic option but many issues are still debated. Further clinical and in vitro studies are needed to clarify the remaining doubts.. However, it’s a very vague statement since they have nether highlighted the “many issues” with MSCs usage nor “remaining doubts” in the field. The authors should make this statement clearer by giving a few examples of both.

2.       The statement by authors, “MSCs are pluripotent progenitor cells derived from a population of adult stem cells that can be isolated from numerous tissues [30].” is wrong and should not be published in its current form. MSCs are multipotent cells but not pluripotent cells. Authors should cite well known papers for this statement.

3.       In the Figure 1,

a.       The authors mention “The Vicious cycle” where they write that “MMPs can produce several fragments of molecules working as DAMPs”, however in the current form of Figure 1 this connection is missing and it is not clear whether MMPs directly produce DAMPs or MMP act on apoptotic cells that generate DAMP. It’s important to highlight that connection here otherwise the vicious cycle remains incomplete..  

b.       Chondrocyite in Fig. 1 should be spelled as Chondrocyte

4.       It’s not clear which phase are the authors referring to in the statement, “Fibronectin is an ECM glycoprotein with high-molecular-weight (~500-~600 kDa) that can be damaged during this phase, causing the production of fibronectin fragments stimulating the chondrolytic process [23].

5.       Recently, “a paper by Lambert et al. highlighted…” should be “a study by Lambert et al. highlighted…”

6.       Line 201: this sentence doesn’t make any sense and should be corrected “…causing a downstream in the inflammatory OA signaling by modulating the…

7.       Line 214: There is an opened small bracket “(” which isn’t closed in the sentence.

8.       This statement doesn’t make sense, “Numerous clinical trials have already reported clinical trials…

9.       Line 321, “MSc” should be “MSCs” and sentence should be in plural, unless it was only one single MSC was derived from adipose tissue which I believe isn’t the case.

10.   The authors have used the term “in the face of” at multiple places in the manuscript and the meaning is sometimes confusing.

Author Response

Reviewer: In the abstract the authors have written “Mesenchymal stem cells (MSCs) may be a promising therapeutic option but many issues are still debated. Further clinical and in vitro studies are needed to clarify the remaining doubts.”. However, it’s a very vague statement since they have nether highlighted the “many issues” with MSCs usage nor “remaining doubts” in the field. The authors should make this statement clearer by giving a few examples of both.

Response: Thanks for the advice. The abstract has been reformulated, indicating some aspects of MSCs that still need to be defined.

Reviewer: The statement by authors, “MSCs are pluripotent progenitor cells derived from a population of adult stem cells that can be isolated from numerous tissues [30].” is wrong and should not be published in its current form. MSCs are multipotent cells but not pluripotent cells. Authors should cite well known papers for this statement.

Response: We thank the reviewer for pointing out this aspect. We corrected the sentence, and we included more recent and appropriate references.

Reviewer: In the Figure 1, the authors mention “The Vicious cycle” where they write that “MMPs can produce several fragments of molecules working as DAMPs”, however in the current form of Figure 1 this connection is missing and it is not clear whether MMPs directly produce DAMPs or MMP act on apoptotic cells that generate DAMP. It’s important to highlight that connection here otherwise the vicious cycle remains incomplete.. 

Chondrocyite in Fig. 1 should be spelled as Chondrocyte

Response: Thank you for the suggestion and sorry for the typo. We uploaded a modified new figure, also correcting it.

Reviewer: It’s not clear which phase are the authors referring to in the statement, “Fibronectin is an ECM glycoprotein with high-molecular-weight (~500-~600 kDa) that can be damaged during this phase, causing the production of fibronectin fragments stimulating the chondrolytic process [23].”

Response: We corrected the sentence by specifying the omitted phase, which refers to that of cartilage degradation and fibronectin proteolytic cleavage.

Reviewer: Recently, “a paper by Lambert et al. highlighted…” should be “a study by Lambert et al. highlighted…”

Response: Thank you for the advice. The sentence has been corrected.

Reviewer: Line 201: this sentence doesn’t make any sense and should be corrected “…causing a downstream in the inflammatory OA signalling by modulating the…”

Response: Thank you for the advice. The sentence has been reformulated.

Reviewer: Line 214: There is an opened small bracket “(” which isn’t closed in the sentence.

Response: Thank you for the advice. The sentence has been corrected.

Reviewer: This statement doesn’t make sense, “Numerous clinical trials have already reported clinical trials…”

Response: Thank you for the advice. The sentence has been corrected.

Reviewer: Line 321, “MSc” should be “MSCs” and sentence should be in plural, unless it was only one single MSC was derived from adipose tissue which I believe isn’t the case.

Response: Thank you for the advice. The sentence has been corrected.

Reviewer: The authors have used the term “in the face of” at multiple places in the manuscript, and the meaning is sometimes confusing.

Response: Thank you for the advice. The use of “in the face of” has been checked and corrected.

Reviewer 3 Report

General comments

The article Knee osteoarthritis: epidemiology, pathogenesis, and mesenchymal stem cells. What else is new? An update is generally well written and well organized. In this review, the authors summarized the current state of knowledge regarding the molecular mechanism of cartilage damage in knee OA, including the latest therapeutic approaches in the field of regenerative medicine. The introduction is far too short, it needs to be thoroughly expanded and the literature needs to be supplemented (currently only 1 item is cited).  The methodology adopted in the paper is not described correctly and needs to be supplemented. The authors present a review paper, and the literature list contains 70 items. In my opinion, this is far too few for this type of article. Conclusions are presented in a concise manner. The work needs significant editorial changes especially in the preparation of the text and the way literature is cited. Please adapt the text to the requirements of the journal. In my opinion, it seems necessary to make major corrections to allow a more thorough understanding of the topic. The following are my comments.

Major comments:

Abstract:

The abstract should contain only the most important information for this review. Please restructure the abstract with a view to reducing repetition. 

Introduction

The introduction is very short, please add more detailed information on degenerative disease with the necessary recent literature. Please emphasize the importance of the problem under consideration and more clearly state the purpose of this review.

The incidence of osteoarthritis is influenced by many factors, such as work, sports participation, musculoskeletal injuries, obesity and gender. Information about this, along with the necessary literature, should be added in the introduction. Authors may find some useful information in the works: DOI 10.1016/S0140-6736(19)30417-9; DOI10.3390/app11041552; https://doi.org/10.1136/annrheumdis-2013-204763; doi:10.1038/nrrheum.2015.135 DOI 10.3390/app10238312; https://doi.org/10.4081/or.2014.5188; doi:10.1038/nrrheum.2015.135;DOI: 10.1056/NEJMcp1903768;

The introduction should be expanded to include more information on typical diagnostic methods (CT, MRI, ultrasound) including physical examination, as well as alternative methods such as vibroarthrography with limitations in the diagnosis of osteoarthritis. This section is important because of the significance of early detection of degenerative changes and enabling the selection of an appropriate therapeutic procedure.   Authors may find some useful information in the works:

https://doi.org/10.1016/j.cpet.2018.08.004; https://doi.org/10.1111/j.1617-0830.2006.00063.x; DOI 10.3390/app9194102; https://doi.org/10.1016/j.berh.2016.09.007; doi:10.35784/acs-2022-14; https://doi.org/10.3390/s22062176; https://doi.org/10.3390/s22103765;

Please make the relevant additions with the necessary literature. This will allow you to better understand the topic and highlight the essence of the issue at hand.

2. Epidemiology of knee osteoarthritis: recent trends

This section is poor in newest knowledge. The topic of obesity was proposed as a risk factor, however, no information concerning proinflammatory factor of obesity was provided. Moreover, risk factors for OA of lower extremity include work related loads, sports, or military training. Malalignment of the lower extremity is also a risk factor. No information is given concerning meniscal tears, and increased loads after meniscal tear. This section even if focused only on knee joint requires to be rewritten and more accurate references provided.

3. Pathogenesis of OA

The importance of superficial cartilage layer in constricting water content in the cartilage, as well as subchondral bone function in development of OA should be discussed in this section. Superficial layer acts as a regulator of water content, injury to superficial layer changes the water content and therefore diminishes load bearing properties of cartilage. Far more information concerning pathogenesis of OA is required in this section including collagen degradation,  increased GAG water intake, calcification of subchondral bone. Moreover, none is written about incapability of chondrocytes to de novo creation of collagen type II. Menisci should be evaluated in regard to fiber orientation and its capacity to load transmission depending on location.

4. The OA inflammatory response: Mechanism and mediators

Relatively good presentation of inflammatory activation molecules.

Materials and Methods

I suggest adding a short Material and Methods. Please provide criteria for article selection, keywords, databases used (e.g. Scopus, WOS, PUBMED), rejection criteria, and justification for the article structure adopted.

After making the appropriate additions, the article may be accepted for publication.

Author Response

Abstract:

Reviewer: The abstract should contain only the most important information for this review. Please restructure the abstract with a view to reducing repetition.

Response: Thanks for the advice. The abstract has been edited and reformulated, indicating some aspects of MSCs that still need to be defined.

Introduction

Reviewer: The introduction is very short, please add more detailed information on degenerative disease with the necessary recent literature. Please emphasize the importance of the problem under consideration and more clearly state the purpose of this review.

Response: we modified the introduction according to your suggestion, we structured it focusing on the current problem (also adding information about hip and knee OA burden), and then we clearly stated the purpose of our paper.

Reviewer: The incidence of osteoarthritis is influenced by many factors, such as work, sports participation, musculoskeletal injuries, obesity and gender. Information about this, along with the necessary literature, should be added in the introduction. Authors may find some useful information in the works: DOI 10.1016/S0140-6736(19)30417-9; DOI: 10.3390/app11041552; https://doi.org/10.1136/annrheumdis-2013-204763; doi:10.1038/nrrheum.2015.135 DOI 10.3390/app10238312; https://doi.org/10.4081/or.2014.5188; doi:10.1038/nrrheum.2015.135; DOI: 10.1056/NEJMcp1903768;

Response: we expanded this section adding the role of several risk factors (sports participation, alignment, obesity and other sociodemographic factors as education, money income). We thank the reviewer for the very interesting papers that we included in our list of references.

Reviewer:  The introduction should be expanded to include more information on typical diagnostic methods (CT, MRI, ultrasound) including physical examination, as well as alternative methods such as vibroarthrography with limitations in the diagnosis of osteoarthritis. This section is important because of the significance of early detection of degenerative changes and enabling the selection of an appropriate therapeutic procedure.  Authors may find some useful information in the works:

https://doi.org/10.1016/j.cpet.2018.08.004; https://doi.org/10.1111/j.1617-0830.2006.00063.x;

DOI 10.3390/app9194102; https://doi.org/10.1016/j.berh.2016.09.007; doi:10.35784/acs-2022-14; https://doi.org/10.3390/s22062176; https://doi.org/10.3390/s22103765;

Response: Thanks for the advice. The introduction has been expanded including more information on diagnostics methods, and the suggested references have been added.

Reviewer: Epidemiology of knee osteoarthritis: recent trends

This section is poor in newest knowledge. The topic of obesity was proposed as a risk factor, however, no information concerning proinflammatory factor of obesity was provided. Moreover, risk factors for OA of lower extremity include work related loads, sports, or military training. Malalignment of the lower extremity is also a risk factor. No information is given concerning meniscal tears, and increased loads after meniscal tear. This section even if focused only on knee joint requires to be rewritten and more accurate references provided.

Response: The Epidemiology section has been extensively expanded according to reviewers’ suggestions.

Reviewer Pathogenesis of OA

The importance of superficial cartilage layer in constricting water content in the cartilage, as well as subchondral bone function in development of OA should be discussed in this section. Superficial layer acts as a regulator of water content, injury to superficial layer changes the water content and therefore diminishes load bearing properties of cartilage. Far more information concerning pathogenesis of OA is required in this section including collagen degradation, increased GAG water intake, calcification of subchondral bone. Moreover, none is written about incapability of chondrocytes to de novo creation of collagen type II. Menisci should be evaluated in regard to fiber orientation and its capacity to load transmission depending on location.

Response: Thank you for the suggestions. We expanded the pathogenesis section, according to the kindly suggested points.

Materials and Methods

Reviewer: I suggest adding a short Material and Methods. Please provide criteria for article selection, keywords, databases used (e.g. Scopus, WOS, PUBMED), rejection criteria, and justification for the article structure adopted.

Response: Thank you for the advice. A short chapter of Material and Methods has been added in the manuscript.

Round 2

Reviewer 1 Report

the authors have done a great deal of work to improve the quality of the article, and have responded to all comments. I recommend the article for printing

Reviewer 3 Report

The article has been significantly supplemented. It can be accepted in its present form.